# Fluorescent SiO_2_@Tb^3+^(PET-TEG)_3_Phen Hybrids as Nucleating Additive for Enhancement of Crystallinity of PET

**DOI:** 10.3390/polym12030568

**Published:** 2020-03-04

**Authors:** Yanna Zhang, Yao Wang, Hao Li, Xuezhong Gong, Jixian Liu, Linjun Huang, Wei Wang, Yanxin Wang, Zhihuan Zhao, Laurence A. Belfiore, Jianguo Tang

**Affiliations:** 1Institute of Hybrid Materials, National Center of International Research for Hybrid Materials Technology, National Base of International Science & Technology Cooperation, College of Materials Science and Engineering, Qingdao University, Qingdao 266071, China; zhangyn720@126.com (Y.Z.); lihao0416178@163.com (H.L.); xzgong@qdu.edu.cn (X.G.); ljx@qdu.edu.cn (J.L.); newboy66@126.com (L.H.); wangwei040901@163.com (W.W.); zhaozhihuande@163.com (Z.Z.); belfiore@engr.colostate.edu (L.A.B.); 2Department of Chemical and Biological Engineering, Colorado State University, Fort Collins, CO 80523, USA

**Keywords:** PET, silica nanoparticles, hybrid polymer, Tb^3+^ complex, fluorescence

## Abstract

A hybrid polymer of SiO_2_@Tb^3+^(poly(ethylene terephthalate)-tetraglycol)_3_ phenanthroline (SiO_2_@Tb^3+^(PET–TEG)_3_Phen) was synthesized by mixing of inorganic SiO_2_ nanoparticles with polymeric segments of PET–TEG, whereas PET–TEG was achieved through multi-step functionalization strategy. Tb^3+^ ions and β-diketonate ligand Phen were added in resulting material. The experimental results demonstrated that it was well blended with PET as a robust additive, and not only promoted the crystallinity, but also possessed excellent luminescence properties. An investigation of the mechanism revealed that the SiO_2_ nanoparticles functioned as a crystallization promotor; the Tb^3+^ acted as the fluorescent centre; and the PET–TEG segments played the role of linker and buffer, providing better compatibility of PET matrix with the inorganic component. This work demonstrated that hybrid polymers are appealing as multifunctional additives in the polymer processing and polymer luminescence field.

## 1. Introduction

Poly(ethylene terephthalate) (PET) is an important industrial material that is widely used for packaging film, fiber, and beverage bottles owing to its excellent thermophysical properties, chemical resistance, and low permeability [1]. However, its slow crystallization rate and poor crystallization [2] affect the properties and further limit its application [3] in industrial production. It is commonly accepted that addition of nucleating additives is one of the most effective protocols to improve the crystallization rate and achieve good crystallinity, which lead to the enhancement of various properties [4,5,6].

Silica nanoparticles (SiO_2_ NPs) with a large surface area have been demonstrated to be nanofillers in the PET matrix, enhancing the mechanical and physical properties owing to the strong interaction between SiO_2_ and the PET chain [7]. Meanwhile, the incompatibility is the big problem in achieving homogeneously dispersed SiO_2_ NPs in the PET matrix. Additionally, specific properties are difficult to integrate into PET, as the bare SiO_2_ lacks functional groups to anchor specific moieties. Notably, hybrid polymer is a special type of nucleating agent [8] that has the below properties of inorganic (stability and rigidity), organics (processability and flexibility), and possessing superior compatibility with the polymer matrix. Furthermore, it provides additional functions (e.g., luminescence, chemical sensing) [9] for the polymer composites [10].

To obtain PET–SiO_2_ hybrids, in situ polymerization by grafting polymer chain [11] onto silica NP is more acceptable because it guarantees the homogeneity of hybridization and avoids the aggregation of silica nanoparticles [12]. In order to control the graft density, the “grafting from” method is usually adopted in the way of introducing initiating groups on the silica surface via in situ polymerization and followed by further polymerization techniques [13]. These protocols also allow luminesce properties for the PET–SiO_2_ hybrids with potential applications in optical fibers, and rare earth elements are excellent candidates in which to be doped. Many researches have reported Eu, Tb, and so on doped polymers obtaining good fluorescent properties [14]. Few reports have focused on the Tb-doped PET–SiO_2_ hybrids. They have great potential to fabricate highly spinnable fluorescent PET optical fiber, and their applications have been extended to the fields of coatings [15,16], membranes [17], OLED [18], and so on.

In this work, the rare-earth doped hybrid polymer of SiO_2_@Tb^3+^(PET–tetraglycol (TEG))_3_Phen was synthesized through the multi-step polymerization process [19]. Successful coating of PET–TEG polymeric blocks and complexation of Tb^3+^ were clearly identified by scanning electron microscopy (SEM) [20], X-ray photoelectron spectroscopy (XPS), and fluorescent spectrometer. The SiO_2_@Tb^3+^(PET–TEG)_3_Phen hybrids were used as nucleating reagents for PET crystallization. The differential scanning calorimeter (DSC) analysis showed that SiO_2_@Tb^3+^(PET–TEG)_3_Phen have a positive effect on enhancing the degree of crystallinity. The fluorescent experiments showed that the SiO_2_@Tb^3+^(PET–TEG)_3_Phen hybrids exhibited good luminescence properties, which give it great potential to fabricate fluorescent PET optical fiber [21].

## 2. Material and Methods

### 2.1. Materials

The commercial PET polyester slices (processing grade: fiber grade, non-extinction, intrinsic viscosity 0.676 dL/g) were purchased from Sinopec Yizheng Chemical Fiber Co., Ltd. (Yizheng, China). The SiO_2_ nanoparticles (amorphous hydrophilicity, particle size is about 100~200 nm) were purchased from Zhejiang Yuda Chemical Co., Ltd. Sulfoxide chloride, benzene, tetrachloroethane, phenol, antimony trioxide, toluene, dimethyl terephthalate (DMT), zinc acetate, acetone, triphenyl phosphite, o-phenanthroline (Phen), and *N*,*N*-dimethylformamide (DMF) were purchased from Sinopharm Chemical Reagent Co., Ltd. (Beijing, China). Tetraethylene glycol (TEG) was obtained from Shanghai Macklin Biochemical Technology Co., Ltd. (Shanghai, China). All the above mentioned reagents are of analytical grade (AR), and were used as received without further purification. Terbium chloride (99.99%) was purchased from Shandong Desheng New Materials Co., Ltd. (Jining, China).

### 2.2. The Preparation of Tb^3+^/PET–TEG/Phen

Firstly, the low molecular weight PET (*M*_n_ = 11,500) was synthesized via condensation polymerization, according to our previous report [22]. Then, phenol and tetrachloroethane (mass ratio 1:1) was mixed with the above PET in a round bottom flask. The mixture was kept in an oil bath of 70 °C for 2 h with continuously stirring. Then, 5 mL ethylene glycol was added dropwise, followed by 30 mg Sb_2_O_3_ and 16 mL TEG. Afterwards, the temperature was raised to 110 °C, and kept for 2.5 h to form PET–TEG. To obtain Tb^3+^/PET–TEG/Phen, an amount of TbCl_3_ (0.02 mol/L) and Phen (0.02 mol/L) were mixed with PET–TEG, and reacted at 70 °C for 6 h.



### 2.3. The Preparation of SiO_2_–TEG

Firstly, SiO_2_ was treated with sulfoxide chloride as follows: 1.5 g of SiO_2_ nanoparticles was dispersed in benzene to form a homogeneous suspension, after which 12.5 mL sulfoxide chloride was added and the mixture was kept at 65 °C for 5 h. The suspension was centrifuged and the precipitate was washed twice with benzene. Then, the solid was dried in an oven (100 °C). Next, 0.5 g of the pre-treated SiO_2_ NPs was dispersed in 20 mL toluene, and mixed with 10 mL tetraethylene glycol for reaction at 70 °C for 5 h with continuously stirring under an N_2_ atmosphere. Finally, the product was washed twice with toluene and dried in an oven.



### 2.4. The Preparation of SiO_2_@Tb^3+^(PET–TEG)_3_Phen

Briefly, 1.4 g of Tb^3+^/PET–TEG/Phen was put into a mixed solution of phenol and tetrachloroethane (mass ratio 1:1) in a round bottom flask, which was placed in an oil bath (70 °C), and 5 mL of ethylene glycol was added. After 1 h, the reaction temperature was raised to 100 °C, following by the addition of 30 mg of Sb_2_O_3_ and 0.4 g SiO_2_–TEG, and the reaction was kept for another 2 h. Afterwards, the solid product was separated and washed twice with acetone and ethanol, respectively. The chemical reaction principle is the same as (1).

### 2.5. The Preparation of PET/SiO_2_@Tb^3+^(PET–TEG)_3_Phen

The prepared SiO_2_@Tb^3+^(PET–TEG)_3_Phen was used as additives to obtain an SiO_2_/Tb-hybridized PET composite. In detail, a different amount of SiO_2_@Tb^3+^(PET-TEG)_3_Phen was blended with commercial PET slices using a micro-blender via the melt blending method. For the different amount of additives, the PET hybrids are named as 0%, 1%, 2%, and 3%.

### 2.6. Characterization

The molecular weight of PET–TEG was characterized by gel permeation chromatography (GPC, HLC-8320GPC, Kyoto, Japan). The column temperature was set at 25 °C, and the column pressure was 31 MPa. The mobile phase was DMF, and the elution time was 1 h. The elements of the prepared additive and PET hybrids were analyzed by X-ray photoelectron spectroscopy (XPS, D8 Advance, Bruker, Cologne, Germany). The excitation source was Mg Kα X-ray, the power was 300 W, and the C1s spectrum (284.8 eV) was used as calibration. The functional groups were analysed by Fourier transform infrared spectroscopy (FTIR, MAGNA-IR 5700, Nicolet, Inc., Palo Alto, CA, USA). The morphology and composition were characterized by transmission electron microscope (TEM, JEM-F200, Kyoto, Japan) and SEM-EDS (Energy Dispersive Spectrometer) (GENESIS XM, Waltham, MA, USA). The fluorescence properties were measured by fluorescence spectrometer (Spectro Xepos, Cologne, Germany) and microspectro phometer (20/30PV, Craic, Inc., Palo Alto, CA, USA). The excitation wavelength was set as 331 nm. The crystallization performance of PET hybrids was characterized by differential scanning calorimeter (DSC, DSC214 Polyma, Cologne, Germany). The temperature range was set from 25 to 280 °C. The ramp rate was 10 °C/min.

## 3. Results and Discussion

### 3.1. Molecular Weight Determination of PET–TEG

As illustrated in Scheme 1, the PET–TEG is a key matrix to integrate SiO_2_ and Tb for the subsequent hybridization with PET slices. It is crucial to determine the molecular weight of PET–TEG in order to hybridize SiO_2_ and Tb with precisely controlled content. As measured by gel permeation chromatography (GPC) [23], the molecular weight distribution curve of PET–TEG shows a single peak, indicating the distribution is narrow. The Mw fraction below 95% located between 17,000 and 70,000 (Appendix A). It can be seen from Table 1 that the LMPET (low molecular weight) sample presented a degree of polydispersity of around 2.5, and the PET–TEG sample presented a degree of polydispersity of around 1.2. In addition, the molecular weights (*M*_w_ and *M*_n_) [24] show a slight variation from the PET sample to PET–TEG; according to statistics, even small differences in molecular weight values can cause considerable differences in PET properties.

### 3.2. Structure Characterization

The synthesis of SiO_2_@Tb^3+^(PET–TEG)_3_Phen required multiple steps, starting from SiO_2_ and PET, and in each step, the obtained samples were characterized by FTIR. (Figure 1) The bands at 794 cm^−1^ and 1058 cm^−1^ are the characteristic stretching vibration absorption peaks of Si–O–Si [25,26], which can be found in both SiO_2_ and SiO_2_–TEG. In SiO_2_–TEG, the C–H (1122 cm^−1^) and C–O (1248 cm^−1^) overlapped with that of Si–O–C and Si–O–Si, which cannot be clearly distinguished. In PET–TEG, the symmetrical vibration absorption peak (721 cm^−1^) [27] can be assigned to the C–O–C group of the PET skeleton. The characteristic absorption peak at 1713 cm^−1^ is ascribed to the C=O of polyester [28]. Compared with PET-TEG, the band at 1650 cm^−1^ of Tb^3+^ (PET–TEG)_3_Phen shows a new absorption peak of the conjugated carbonyl; it can be assigned to the incorporation of the Tb-complex [29], in which Tb coordinated with nitrogen and oxygen atoms, as shown in Scheme 1. In SiO_2_@Tb^3+^(PET–TEG)_3_Phen, the presence of the bands at 1058 cm^−1^ and 1650 cm^−1^ confirmed the existence of SiO_2_ and Tb^3+^, respectively, indicating the synthetic protocol was well conducted and the final samples were successfully prepared.

To obtain bond information of the samples, XPS was used to analyse the C-bond and the results are shown in Figure 2. In the deconvolution spectrum of C1s in SiO_2_–TEG, the peaks at 284.4 and 286.1 eV can be assigned to C–C and C–O–C, respectively [29], verifying the grafting of TEG moieties onto SiO_2_. Further, the increased C content (12.74%), compared with SiO_2_, can also prove this (Table 2). In the sample of SiO_2_@Tb^3+^ (PET–TEG)_3_Phen, the C=C peak was deconvoluted at 284.5 eV, which can be from the benzene ring of Phen [30]. The peak that appeared at 288.6 eV can be assigned to the O=C=O, indicating the presence of PET. Element content analysis (Table 2) also proved the presence of Tb (0.79%, atomic ratio) and N (comes from Phen). These results demonstrated the successful synthesis of SiO_2_@Tb^3+^ (PET–TEG)_3_Phen.

### 3.3. Characterization of Fluorescence Properties

The fluorescence properties of the synthesized Tb^3+^/PET–TEG/Phen and SiO_2_@Tb^3+^(PET–TEG)_3_Phen were measured by fluorescence spectrometer, and the results are shown in Figure 3. The excitation wavelength was determined at 331 nm (Figure 3b). In Figure 3a, the fluorescence intensity of SiO_2_@Tb^3+^(PET–TEG)_3_Phen is much weaker compared with Tb^3+^/PET–TEG/Phen. It is believed that the Tb^3+^/PET–TEG/Phen was grafted onto the SiO_2_ surface through crosslink. Thus, the SiO_2_ play the role of collectors, which accumulated surrounding Tb^3+^ and increased the local concentration of PET–TEG segments. In this case, much of the unfolded PET–TEG segments would lead to the decrease in fluorescence intensity. To verify this hypothesis, Tb^3+^/PET–TEG/Phen with different concentrations was prepared in DMF from 0.002 mol/L to 0.01 mol/L to investigate the fluorescence behaviour. It can be seen that there are four fluorescence emission peaks [31], with the strongest fluorescence emission peak being the ^5^D_4_–^7^F_5_ transition. With the increasing concentration, the fluorescence intensity increased. However, the intensity reached the maximum at the concentration of 0.006 mol/L, where a further increase in concentration leads to fluorescence fading. Limited solubility of PET–TEG segments in DMF induced the excess of folded PET–TEG segments, which caused Tb^3+^ fluorescence quenching. These results also implied the potential usage of SiO_2_@Tb^3+^(PET–TEG)_3_Phen not only in crystallization enhancement, but also in fluorescent polymers.

### 3.4. Morphological Characterization

Figure 4 showed the TEM images and the elements content of SiO_2_@Tb^3+^(PET–TEG)_3_Phen. The SiO_2_ is monodisperse with an average size of around 150 nm. The nanoparticles are discrete without connection, indicating a clean surface (Figure 4a). For comparison, the SiO_2_–TEG exhibited a coating outside the SiO_2_ surface and crosslinked with each other (Figure 4b). This solidly proved that the TEG-moieties successfully grafted onto SiO_2_. As shown in Figure 4c, the SiO_2_ sphere turned to be more irregular, indicating the Tb^3+^/PET–TEG/Phen moieties had been grafted. SEM-EDS and elemental analysis demonstrated the presence of C, O, Si, and Tb (Figure 4d, Appendix A), and further proved the successful synthesis of SiO_2_@Tb^3+^(PET–TEG)_3_Phen. The content of Si and Tb was 2.02 and 0.01 wt %, respectively, corresponding to SiO_2_ and TbCl_3_ (Table 3), which is in accordance with the experimental operation of SiO_2_–PET–TEG and Tb^3+^/PET–TEG/Phen.

Figure 5 shows the high-magnification field emission scanning electron microscopy (FESEM) images; from the figure, we can see that the nano-SiO_2_ aggregates more seriously in PET, the size of the agglomerates is not uniform, and the average size is more than 250 nm. After modification, its dispersion in PET increases and agglomeration decreases [32]. In addition, TEG–PET is grafted on the surface of SiO_2_, making the nucleating agent more compatible with the PET matrix, and the size of the part of the nucleating agent dispersed in PET is 100–150 nm.

### 3.5. Crystallization Behaviour of PET/SiO_2_@Tb^3+^(PET–TEG)_3_ Phen Hybrid

The synthesized SiO_2_@Tb^3+^(PET–TEG)_3_Phen was used as an additive to blend with PET, obtaining SiO_2_/Tb–PET hybrids. The crystallization of PET samples was measured with differential scanning calorimeter (DSC). The effect of SiO_2_@Tb^3+^(PET–TEG)_3_Phen on the crystallization of PET can be characterized by the cold-crystallization temperature (*T*_cc_) and melt-crystallization temperature (Tmc) of PET samples, where lower *T*_cc_ or higher *T*_mc_ corresponds to the higher crystallization rate.

Figure 6 shows the crystallization behavior of pristine PET, as well as that with different amounts of additives, namely 1%, 2%, and 3% of SiO_2_@Tb^3+^(PET–TEG)_3_Phen. Obviously, the *T*_mc_ of PET increased greatly with the addition of SiO_2_@PET–TEG–PET (2%) and SiO_2_@Tb^3+^(PET–TEG)_3_Phen (2%) (Appendix A). According to the calculated crystallinity (Table 4), it can be found that, compared with SiO_2_, the addition of Tb^3+^ in the SiO_2_@Tb^3+^(PET–TEG)_3_Phen did not contribute too much in the crystallization of PET hybrids, indicating that the SiO_2_ played the significantly effect to enhance the crystallinity.

The presence of the benzene ring in the main chain structure of the polymer molecule made it unfavorable for molecule chain movement, which led to a low crystallization rate and poor crystallinity [33,34]. With the addition of SiO_2_@Tb^3+^(PET–TEG)_3_Phen as additives, the polymer segments that grafted on the surface of SiO_2_ afford SiO_2_ possess excellent compatibility with the PET matrix and resulted in a good dispersibility [35]. Taking advantage of that, the internal structure of PET polymer was changed and the chain mobility was improved. Thus, the SiO_2_@Tb^3+^(PET–TEG)_3_Phen act as a nucleating agent, and promoted the crystallization rate and crystallinity of PET. Most importantly, the incorporated Tb^3+^ affords the PET hybrid excellent fluorescent properties with light emission at 550 nm, which can be processed to be fluorescent fiber and film in the fields of textile and capacitors [36].

### 3.6. Effects on Environment

With the wide spread application, PET waste disposal poses a serious problem to maintaining a clean environment [37]. However, it should be pointed out that PET does not create a direct hazard to the environment, but because of its substantial fraction by volume in the waste stream and its high resistance to the atmospheric and biological agents, it is seen as a noxious material.

To ease the threat to environment, PET recycling seems to be the best way for environmental remediation. As one of the world’s most recyclable polymer [38], PET with functional ester groups can be cleaved by some reagents, such as water (hydrolysis), alcohols (alcoholysis), acids (acidolysis), glycols (glycolysis), and amines (aminolysis). The polymer backbone is degraded into monomer units (i.e., depolymerisation) or randomly ruptured into larger chain fragments (i.e., random chain scission) with associated formation of gaseous products. Therefore, monomers, petroleum liquids, and gases can be yielded through chemical recycling, and furthermore, monomers are purified by distillation and drying and used for manufacture of polymers.

## 4. Conclusions

In summary, inorganic SiO_2_ nanoparticles and rare earth Tb^3+^ ions were effectively linked with PET–TEG polymeric segments to form a hybrid polymer through a systematic strategy. Fluorescence characterization showed good fluorescence properties of the hybrid polymer and the quench mechanism was investigated. As an additive to PET, the SiO_2_@Tb^3+^(PET–TEG)_3_Phen hybrid polymer was well blended with PET matrix to improve the *T*_c_ and *T*_m_ by 5.7% and 7.3%, respectively, with respect to pure PET. The improved crystallinity was ascribed to the portion of SiO_2_ nanoparticles, which function as nucleation sites. The Tb^3+^ acted as the fluorescent centre, and the PET–TEG segments played the role of linker and buffer to contribute better dispersibility of SiO_2_ nanoparticles in the PET matrix. Overall, this work paves a way for the synthesis of multifunctional polymer hybrids to meet the needs in industrial and biomedical fields such as display screens and drug delivery indicator.

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
