# Peer review of "Fluorescent SiO2@Tb3+(PET-TEG)3Phen Hybrids as Nucleating Additive for Enhancement of Crystallinity of PET"

_polymers, 2020, doi:10.3390/polym12030568_

Round 1

Reviewer 1 Report

The present manuscript entitled “Fluorescent SiO2@Tb3+(PET-TEG)3Phen Hybrids for Enhancement of Crystallinity of PET, polymers-683915" could not be accepted for its publication in its present format. A new version of the manuscript, with a moderate revision, which included the following recommendation, that the authors must take into account in the revised version, Only after these modifications could be considered for their publication in Polymers (MDPI):

1) First of then, a general comment about the submitted manuscript.
- The authors present a general procedure for the synthesis of a hybrid polymer with an organic composition (PET-TEG) ​​doped with SiO2 and Tb (III) that gives the polymer adequate characteristics regarding its structural (crystallinity) and optical (luminescent) properties. , among others. However, at no time stand out the capabilities and use of these characteristics in the material once formed and its potential added value. In any case, they do not highlight it correctly.
- Given the importance of the synthesized polymers and their toxicity as waste microplastics and their environmental pollution. All research conducted in this way must incorporate a section that explaining as clearly as possible the potential degradation and environmental remediation pathways.

2) It is suggested that the title of the work be changed in order to include the concept of polymer multifunctionality.
3) Also, It is suggested that replacing the term "organic-inorganic" about the polymer, by "hybrid polymer". This term appears in the abstract and also throughout the whole manuscript.
4) The techniques for characterizing the structural and luminescent properties of the new hybrid polymer are suitable. However, they should be commented on with more significant differentiation between them. Its potential should also be commented on its industrial utility.
5) The industrial, commercial utility, or the "added value" of this new hybrid polymer should be explained with concrete examples.
6) As indicated above, guidelines for degradation or remediation of the polymer should be included.
5. The Conclusion must rewrite according to the previous discussion established in the new version.

Author Response

Dear Reviewer:

We would like to thank you for your kind and valuable comments on our manuscript entitled ‘Fluorescent SiO2@Tb3+(PET-TEG)3Phen Hybrids for Enhancement of Crystallinity of PET’ (Manuscript ID: polymers-683915, Type of manuscript: Article), which are truly helpful for us to revise and improve our manuscript. In the revised manuscript, the changes made in response to these comments have been highlighted in red. Meanwhile, the comments have been carefully addressed item-by-item as listed below. We do hope this revised manuscript could meet the requirements for Polymers.

With best regards

Sincerely yours,

Yao Wang

Reviewer’s comments:

Reviewer #1:

The present manuscript entitled “Fluorescent SiO2@Tb3+(PET-TEG)3Phen Hybrids for Enhancement of Crystallinity of PET, polymers-683915" could not be accepted for its publication in its present format. A new version of the manuscript, with a moderate revision, which included the following recommendation, that the authors must take into account in the revised version, Only after these modifications could be considered for their publication in Polymers (MDPI):

First of then, a general comment about the submitted manuscript.
- The authors present a general procedure for the synthesis of a hybrid polymer with an organic composition (PET-TEG) ​​doped with SiO2 and Tb (III) that gives the polymer adequate characteristics regarding its structural (crystallinity) and optical (luminescent) properties. , among others. However, at no time stand out the capabilities and use of these characteristics in the material once formed and its potential added value. In any case, they do not highlight it correctly.
- Given the importance of the synthesized polymers and their toxicity as waste microplastics and their environmental pollution. All research conducted in this way must incorporate a section that explaining as clearly as possible the potential degradation and environmental remediation pathways.

Response: Thanks for the kind suggestions. We modified it based on your suggestions

Polyethylene terephthalate has wide spread application, on the other hand, rare earth complexes have excellent luminescent properties , but the applications are limited due to their poor thermal stability. According to research, the introduction of rare earth complexes into the inorganic matrix can not only improve its thermal stability, but also improving its luminescent performance, thereby protect it from the external environment such as moisture. The hybrid polymer of SiO2@Tb3+(PET-TEG)3Phen has good compatibility, it can be used as a functional additive to PET. If it can be made into luminescent PET, it can be widely used in the fields of building decoration, transportation and night work, for example, this kind of luminescent fiber can be  applied to fire-fighting clothing, providing an extra guarantee for police safety.

However, the wide spread application will inevitably cause environmental pollution, recycle and reprocess are the key to make up for the deficiency. There are four categories of polymer recycle, including primary, secondary, tertiary and quaternary recycling. Primary recycling refers to pre-consumer industrial scrap, recycled waste is mixed with raw materials to ensure product quality or used as secondary materials; Secondary recycling is mechanical recycling, in this way, the polymer is separated from the contaminants and then reprocessed into granules by conventional melt extrusion; Tertiary recycling is chemical recycling, it involves transformation of polymer chain and poses degradation of polymer backbone into monomer units during process; And quaternary recycling refers to energy recovery, which is what we usually refer to as incineration. In addition, researchers are constantly working on more effective solutions.

It is suggested that the title of the work be changed in order to include the concept of polymer multifunctionality.

Response: Thanks for the reviewer’s good suggestion. The title has been updated as ‘A Fluorescent SiO2@Tb3+(PET-TEG)3Phen Nucleating Additive for Enhancement of Crystallinity of PET’.

Also, It is suggested that replacing the term "organic-inorganic" about the polymer, by "hybrid polymer". This term appears in the abstract and also throughout the whole manuscript.

Response: Thanks for the kind recommendation. The relevant terms have been corrected in the revised manuscript.

The techniques for characterizing the structural and luminescent properties of the new hybrid polymer are suitable. However, they should be commented on with more significant differentiation between them. Its potential should also be commented on its industrial utility.

Response: Thanks for the good recommendation. Compare with PET-TEG, the band at 1650 cm-1 of Tb3+(PET-TEG)3Phen shows a new absorption peak of the conjugated carbonyl, it can be assigned to the incorporation of Tb-complex;

In Figure 3 (a), the fluorescence intensity of SiO2@Tb3+(PET-TEG)3Phen is much weaker compare with Tb3+/PET-TEG/Phen. It is believed that the Tb3+/PET-TEG/Phen was grafted onto the SiO2 surface through crosslink.

Polyethylene terephthalate has wide spread application, it is usually used in synthetic fibers and is a thermoplastic polymer resin of the polyester family that, it is one of the most important raw materials used in man-made fibers. In addition, it also used for microwave food trays and food packaging films.

The industrial, commercial utility, or the "added value" of this new hybrid polymer should be explained with concrete examples. 

Response: Thanks for the good recommendation. As said above, luminescent PET can be widely used in the fields of building decoration, transportation and night work, for example, this kind of luminescent fiber can be applied to fire-fighting clothing, providing an extra guarantee for police safety.

As indicated above, guidelines for degradation or remediation of the polymer should be included.

Response: Thanks for the good recommendation. The guidelines for degradation or remediation of the polymer is as following,

 However, the wide spread application will inevitably cause environmental pollution, recycle and reprocess are the key to make up for the deficiency. There are four categories of polymer recycle, including primary, secondary, tertiary and quaternary recycling. Primary recycling refers to pre-consumer industrial scrap, recycled waste is mixed with raw materials to ensure product quality or used as secondary materials; Secondary recycling is mechanical recycling, in this way, the polymer is separated from the contaminants and then reprocessed into granules by conventional melt extrusion; Tertiary recycling is chemical recycling, it involves transformation of polymer chain and poses degradation of polymer backbone into monomer units during process; And quaternary recycling refers to energy recovery, which is what we usually refer to as incineration. In addition, researchers are constantly working on more effective solutions.

   We are conducting experiments to further optimize the processing and prepare solution to degrade waste PET by chemical recycling, I believe that this problem can be solved well in the future.

The Conclusion must rewrite according to the previous discussion established in the new version.

Response: Thanks for the good recommendation. The conclusion has been updated in the revised manuscript.

In summary, the inorganic SiO2 nanoparticles and rare earth Tb3+ ions are effectively linked with PET-TEG polymeric segments to form a hybrid polymer material through a systematic strategy. As an additive of the synthesized SiO2@Tb3+(PET-TEG)3Phen, it is well blended with PET to obtain modified PET polymer, which the crystallinity is greatly improved. In addition, rare earth complexes have excellent luminescent properties, but the applications are limited due to their poor thermal stability, the introduction of rare earth complexes into the inorganic matrix can improve the stability and possessed excellent luminescence properties. Investigation on the mechanism revealed the SiO2 nanoparticles function as nucleation sites, promoting the crystallization rate. The Tb3+ acted as the fluorescent centre, and the PET-TEG segments played the role of linker and buffer to contribute better dispersibility of SiO2 nanoparticles in the PET matrix. Overall, the luminescent PET can be widely used in the fields of building decoration, transportation and night work, for example, it can be applied to fire-fighting clothing, providing an extra guarantee for police safety.

Author Response

Dear reviewer:

We would like to thank you for your kind and valuable comments on our manuscript entitled ‘Fluorescent SiO2@Tb3+(PET-TEG)3Phen Hybrids for Enhancement of Crystallinity of PET’ (Manuscript ID: polymers-683915, Type of manuscript: Article), which are truly helpful for us to revise and improve our manuscript. In the revised manuscript, the changes made in response to these comments have been highlighted in red. Meanwhile, the comments have been carefully addressed item-by-item as listed below. We do hope this revised manuscript could meet the requirements for Polymers.

With best regards

Sincerely yours,

Yao Wang

Reviewer’s comments:

Reviewer #2:

The MS presents the results of an experimental study aimed to produce PET formulations with enhanced crystallinity and exhibiting fluorescence properties through the addition of small amounts of newly prepared additive, which is referred to as SiO2-Tb3+(PETTEG)3Phen hybrid.

The comments on my evaluation of the manuscript are as follows:

Title: - It is difficult to see why the term hybrid has been used for the additive used as it has been produced from “preformed” silica nanoparticles rather than being “generated insitu” by the sol gel method.

Response: Thanks for the good comments. Firstly, the title has been updated as ‘A Fluorescent SiO2@Tb3+(PET-TEG)3Phen Nucleating Additive for Enhancement of Crystallinity of PET’. And, ‘hybrid’ was used is due to the whole preparation process of SiO2@Tb3+(PET-TEG)3Phen.

Abstract: - a) Abbreviations and acronyms are used without describing the nature of the materials in full. b) The grammar in the first sentence is not correct.

Response: Thanks for the good comments. a) We have reedited the abbreviations and acronyms that appear for the first time in abstract, the changes are as follows: ‘SiO2@Tb3+(Poly(ethylene terephthalate)-Tetraglycol)3Phenanthroline (SiO2@Tb3+(PET-TEG)3Phen)’.

b) The first sentence has been edited by a native English speaking colleague, it is as follows:

‘Hybrid polymer of SiO2@Tb3+(Poly(ethylene terephthalate)-Tetraglycol)3 Phenanthroline (SiO2@Tb3+(PET-TEG)3Phen) was synthesized by mixing of inorganic SiO2 nanoparticles with polymeric segments of PET-TEG, whereas PET-TEG was achieved through multi-step functionalization strategy. Tb3+ ions and β-diketonate ligand Phen was added in resulting material’.

Introduction: - a) Lines 36-39. There are contradictory statements regarding SiO2/PET interactions (strong) and poor dispersion (due to incompatibility?) b) Grammar in lines 39-43 requires considerable attention.

Response: Thanks for the good suggestion and this is a good question. a) SiO2 has high surface activity is due to unsaturated residual bonds on the surface of nano-SiO2, especially the existence of hydroxide on the surface, lots of particles connect by hydrogen bond and form branched chains, and the branched chains interact with each other by hydrogen bonds to form a three-dimensional chain structure, thereby forming secondary particle, even the coacervate. Thus, there are contradictory statements regarding SiO2/PET interactions (strong) and poor dispersion.

b) The sentence has been recorrected in lines 42-46, as following,

‘Additionally, specific properties are difficult to be integrated into PET since the bare SiO2 is lack of functional groups to anchor specific moieties. Notably, hybrid polymer is a special type of nucleating agent [8] which emerges the given below properties of  inorganic (stability and rigidity), organics (processability and flexibility) and possessing superior compatibility with polymer matrix’.

Methods and Materials: - a) The choice of the extremely complex systems used for the intended purpose has not been adequately explained and justified. b) The rationale for each step of the procedure used are not explained in terms of expected chemical reactions. c) The “schematic route”, Scheme 1, is very speculative and not sufficiently clear.

Response: Thanks for the good recommendation. a) The main system contents are as follows: 1) surface modification of SiO2 grafted with TEG, the structure characterization is tested by FTIR, and morphology characterization is tested by TEM; 2) Block copolymer PET-TEG was grafted by TEG and LMPE, its structure was characterized by FTIR, the molecular weight was determined by GPC, and then rare earth complexes were complexed with Tb and phen, test their fluorescence properties. 3) the complexes obtained above were reacted with modified SiO2 to obtain SiO2@Tb3+(PET-TEG)3Phen, which was characterized by FTIR, XPS, EDS, TEM and SEM.

b) We have added the chemical reaction principle of each step to the article and modified, it is as follows:

c) The “schematic route”, Scheme 1 has been updated in article, as follows: Results and Discussion: - a) There are no data for monitoring reactions taking place in the preparation steps of the additive. b) The TEM images in Fig 4 are not sufficiently explanatory to demonstrate the formation of a “hybrid” and there is no micrograph to show the state of dispersion in the PET matrix. c) Data for MW analysis are not compared in detail with the original PET and no explanation has been given for the possible reactions and type of structures responsible for the difference. d) In Table S1 the Tm for PET is given as 242 °C and in Table 4 is reported as 244 °C, whereas all the data in the literature indicate that the true value is around 260 °C. e) The authors state that the confirmation of the structure for the Tb complex by FTIR is in accordance with data published in Ref 28. Neither the title nor the Abstract of the paper for Ref 28 makes any mention of Tb complexes. f) Authors claim that the additive has a strong nucleation effect of the crystallization of PET. While the data show that the degree of crystallinity has hardly been affects and that there is no difference on Tc values between SiO2-PET-TEG and SiO2-Tb3+(PET-TEG)3Phen.

Response: Thanks for the professional suggestion. a) Each step in the preparation process of SiO2-Tb3+(PET-TEG)3Phen has undergone monitoring. Its structure has been analyzed by FTIR, XPS, XRD, etc; Its morphology has been analyzed by TEM, SEM, etc; Its molecular weight has been analyzed by GPC.

b) We re-prepared the samples and the test results have been added in Figure 5 in the revised manuscript, as following,

Figure 5. FESEM images of composites (a) PET/SiO2; (b) PET/SiO2-TEG-PET

Figure 5 showed the high-magnification field emission scanning electron microscopy(FESEM) images, from the figure we can see that the nano-SiO2 aggregates more seriously in PET, the size of the agglomerates is not uniform, and the average size is more than 250 nm. After modification, its dispersion in PET increases and agglomeration decreases [31]. In addition, TEG-PET is grafted on the surface of SiO2, making the nucleating agent more compatible with the PET matrix, and size of the part of the nucleating agent dispersed in PET is 100-150nm.

Antoniadis, G.; Paraskevopoulos, K.M.; Bikiaris, D.; Chrissafis, K. Non-isothermal crystallization kinetic of poly (ethylene terephthalate)/fumed silica (PET/SiO2) prepared by in situ polymerization. Thermochim. Acta 2010, 510, 103-112. c) We re-prepared the samples of LMPET and re-conducted the molecular weight, and the test results have been updated in Table 1 in the revised manuscript and the related analysis has been added to the manuscript, as following,

 It can be seen from Table 1 that LMPET sample presented a degree of polydispersity around 2.5, and PET-TEG sample presented a degree of polydispersity around 1.2. In addition, the molecular weights (Mw and Mn)[27] show a slight variation from PET sample to PET-TEG, according to statistics, even the small differences in molecular weight values can cause considerable differences in PET properties.

Sanches, N.B.; Dias, M.L.; Pacheco, E. Comparative techniques for molecular weight evaluation of poly (ethylene terephthalate)(PET). Polym. Test. 2005, 24, 688-693.

Table 1

Sample

Mn

Mw

Mw/Mn

LMPET

11500

29095

2.531

PET-TEG

24520

30130

1.229

 The structural formulas of PET and PET-TEG are as follows.

There are many types of grafting from the structural formulas, so the results of molecular weight are different. The polydispersity(MW/Mn) of LMPET is higher than PET-TEG, so the latter is more polydisperse.

d) we have retested the DSC of PET by several times, the results as follows. The Tm is slightly higher than before, but it is still lower than 260℃, we speculated that the slight difference is due to the difference between the two samples of PET. The commercial PET polyester slices (Processing grade: Fiber grade, non-extinction, intrinsic viscosity 0.676 dL/g) was purchased from Sinopec Yizheng Chemical Fiber Co., Ltd. The results which we have tested are always lower than 260℃,it is possible that the purity is not enough, affecting the crystallinity. e) We compare the pink line with the green, there is a slight deference between them. the band at 1650 cm-1 can be assigned to the Tb-complex. There are characteristic absorption peaks of C=O at 1713 cm-1 in both Ref 28 and this article. And then, new absorption peak appears due to the incorporation of the Tb-complex, it shows the absorption peak of the conjugated carbonyl.

f) The TEG segment increases the free volume of LMPET-TEG, making it easier for LMPET molecules to enter the crystal lattice. With the increase of the length of the ethoxylated surface grafted on the SiO2, the interaction between the SiO2 and PET molecular chain increases a lot, which is beneficial to the winding of the PET molecular chain on the surface of the nucleating agent. In addition, as a flexible chain, the ethoxy group is conducive to the movement of the nucleating agent, making PET molecules more likely to adhere to the surface of the nucleating agent and nucleate. The formed nucleus induces the crystallization of PET, the Tc and Tm Xc have increased significantly. We will replace the ‘strong nucleation effect of the crystallization’ with a slight improvement, making it more rigorous, and the work to further optimize its crystallinity is ongoing. The Tb3+ just acted as the fluorescent centre, it just gives the nucleating agent luminous properties, and increases the application range, and has no effect on its crystallinity. Conclusions: - The values of the data obtained are grossly overstated in the last sentence. The authors should have cited at least one example of possible fields of application.

Response: Thanks for the kind suggestion. The conclusion has been updated in the revised manuscript, it is as follow:

‘In summary, the inorganic SiO2 nanoparticles and rare earth Tb3+ ions are effectively linked with PET-TEG polymeric segments to form a hybrid polymer material through a systematic strategy. As an additive of the synthesized SiO2@Tb3+(PET-TEG)3Phen, it is well blended with PET to obtain modified PET polymer, which the crystallinity is greatly improved. In addition, rare earth complexes have excellent luminescent properties, but the applications are limited due to their poor thermal stability, the introduction of rare earth complexes into the inorganic matrix can improve the stability and possessed excellent luminescence properties. Investigation on the mechanism revealed the SiO2 nanoparticles function as nucleation sites, promoting the crystallization rate. The Tb3+ acted as the fluorescent centre, and the PET-TEG segments played the role of linker and buffer to contribute better dispersibility of SiO2 nanoparticles in the PET matrix. Overall, the luminescent PET can be widely used in the fields of building decoration, transportation and night work, for example, it can be applied to fire-fighting clothing, providing an extra guarantee for police safety.

Round 2

Reviewer 1 Report

Comment only to the Editor.

Author Response

Response to Reviewer 1 Comments

Point 1: The First of then, a general comment about the submitted manuscript.
- The authors present a general procedure for the synthesis of a hybrid polymer with an organic composition (PET-TEG) ​​doped with SiO2 and Tb (III) that gives the polymer adequate characteristics regarding its structural (crystallinity) and optical (luminescent) properties. , among others. However, at no time stand out the capabilities and use of these characteristics in the material once formed and its potential added value. In any case, they do not highlight it correctly.
- Given the importance of the synthesized polymers and their toxicity as waste microplastics and their environmental pollution. All research conducted in this way must incorporate a section that explaining as clearly as possible the potential degradation and environmental remediation pathways.

Response 1: We appreciated the reviewer’s comments. We highlighted the potential value in the revised manuscript. We also include a paragraph to discuss the concern of PET polymer toxicity and remediation, as below:  

“With the wide spread application, the PET waste disposal poses a serious problem to maintain a clean environment. However, it should be pointed out, that PET does not create a direct hazard to the environment, but due to its substantial fraction by volume in the waste stream and its high resistance to the atmospheric and biological agents, it is seen as a noxious material.

To ease the threaten to environment, PET recycling seems to be the best way for environmental remediation. As one of the world’s most recyclable polymer, PET with functional ester groups that can be cleaved by some reagents, such as water (hydrolysis), alcohols (alcoholysis), acids (acidolysis), glycols (glycolysis), and amines (aminolysis). The polymer backbone is degraded into monomer units (i.e. depolymerisation) or randomly ruptured into larger chain fragments (i.e. random chain scission) with associated formation of gaseous products. Therefore, monomers, petroleum liquids and gases can be yield through chemical recycling, and furthermore, monomers are purified by distillation and drying, and used for manufacture of polymers.”

Point 2: It is suggested that the title of the work be changed in order to include the concept of polymer multifunctionality.

Response 2: We appreciated the reviewer’s good suggestion. The title has been updated as ‘Fluorescent SiO2@Tb3+(PET-TEG)3Phen as Nucleating Additive for Enhancement of Crystallinity of PET’.

Point 3: Also, It is suggested that replacing the term "organic-inorganic" about the polymer, by "hybrid polymer". This term appears in the abstract and also throughout the whole manuscript.

Response 3: We are grateful to the reviewer’s comments. The relevant terms have been corrected and highlighted in the revised manuscript.

Point 4: The techniques for characterizing the structural and luminescent properties of the new hybrid polymer are suitable. However, they should be commented on with more significant differentiation between them. Its potential should also be commented on its industrial utility.

Response 4: We appreciated the reviewer’s comments. We have made corrections and supplementary in the revised manuscript. We involved some discussions on the potential industrial utility as below:

“Given the combination of fluorescence and enhanced crystallinity, the synthesized SiO2@Tb3+(PET-TEG)3Phen hybrid polymer can be the materials for processing of fluorescent fibres, films, foams, sheets, etc, which is potentially applied as flexible screens, displays for industry and drug deliver indicator for biomedical application.”

Point 5: The industrial, commercial utility, or the "added value" of this new hybrid polymer should be explained with concrete examples. 

Response 5: We appreciated the reviewer’s comments. As discussed about the multifunctionallity, for example, SiO2@Tb3+(PET-TEG)3Phen microsphere can be used as indicator for drug deliver for biomedical application due to the excellent biocompatibility of SiO2. The polymeric chain can be grafted to drug deliver. Moreover, the enhanced crystallinity and fluorescence afford the microsphere both mechanical strength and label function. Thus the drug delivery process can be easily monitored by fluorescent microscopy, which provide an alternative solution for advanced medical therapy.

Point 6: As indicated above, guidelines for degradation or remediation of the polymer should be included.

Response 6: We appreciated the reviewer’s comments. we have corrected in the revised manuscript.

Point 7: The Conclusion must rewrite according to the previous discussion established in the new version.

Response 7: We appreciated the reviewer’s comments. The updated conclusion section is below and also highlighted in the revised manuscript.

“In summary, the inorganic SiO2 nanoparticles and rare earth Tb3+ ions was effectively linked with PET-TEG polymeric segments to form a hybrid polymer through a systematic strategy. Fluorescent characterization showed good fluorescent properties of the hybrid polymer and the quench mechanism is investigated. As an additive to PET, the SiO2@Tb3+(PET-TEG)3Phen hybrid polymer was well blended with PET matrix to improve the Tc and Tm by 5.7% and 7.3%, respectively, with respect to pure PET. The improved crystallinity was ascribed to the portion of SiO2 nanoparticles, which function as nucleation sites. The Tb3+ acted as the fluorescent centre, and the PET-TEG segments played the role of linker and buffer to contribute better dispersibility of SiO2 nanoparticles in the PET matrix. Overall, this work pave a way for synthesis of multifunctional polymer hybrids to meet the needs in industrial and biomedical fields such as display screens and drug delivery indicator.”

Reviewer 2 Report

The main problem with this paper is still the " English grammar". I am sure that some of the authors should be able to make the required improvements. Without such corrections the paper will lose its appreciation by the readers.

Although the reaction schemes are realistic and nicely presented, there is still a lack of information about the yields of the products.

There are also still some incoherence, such as the statements in Section 1, lines 73-82. These have nothing to do with the contents of the paper.

The claims about the beneficial effects of this so-called "hybrid" on crystallinity of the products is overstated (from 30.72 %  to 33.04 %). 

Author Response

Response to Reviewer 2 Comments

Point 1: The main problem with this paper is still the " English grammar". I am sure that some of the authors should be able to make the required improvements. Without such corrections the paper will lose its appreciation by the readers.

Response 1: Thanks for the kind comments. The article has been edited by some authors, we hope to get your approval.

Point 2: Although the reaction schemes are realistic and nicely presented, there is still a lack of information about the yields of the products

Response 2: Thanks for the good recommendation and you are very thoughtful. We regret that we cannot measure quantum yield due to limited conditions now.

Point 3: There are also still some incoherence, such as the statements in Section 1, lines 73-82. These have nothing to do with the contents of the paper.

Response 3: Thanks for your helpful recommendation. The reviewer#1 suggests that all research conducted in this way must incorporate a section that explaining as clearly as possible the potential degradation and environmental remediation pathways. We removed this section based on your suggestions, and added the 3.5 Effects to environment in the revised manuscript.

As bellow:

With the wide spread application, the PET waste disposal poses a serious problem to maintain a clean environment. However, it should be pointed out, that PET does not create a direct hazard to the environment, but due to its substantial fraction by volume in the waste stream and its high resistance to the atmospheric and biological agents, it is seen as a noxious material.

To ease the threaten to environment, PET recycling seems to be the best way for environmental remediation. As one of the world’s most recyclable polymer, PET with functional ester groups that can be cleaved by some reagents, such as water (hydrolysis), alcohols (alcoholysis), acids (acidolysis), glycols (glycolysis), and amines (aminolysis). The polymer backbone is degraded into monomer units (i.e. depolymerisation) or randomly ruptured into larger chain fragments (i.e. random chain scission) with associated formation of gaseous products. Therefore, monomers, petroleum liquids and gases can be yield through chemical recycling, and furthermore, monomers are purified by distillation and drying, and used for manufacture of polymers.

Point 4: The claims about the beneficial effects of this so-called "hybrid" on crystallinity of the products is overstated (from 30.72 % to 33.04 %).

Response 4: Thanks for your kind comments. Regarding the exaggeration of crystallinity, we have made appropriate adjustments in the conclusion section in the revised manuscript, and in the next experiment we will work hard to make its crystallinity greatly improved.

As below:

In summary, the inorganic SiO2 nanoparticles and rare earth Tb3+ ions was effectively linked with PET-TEG polymeric segments to form a hybrid polymer through a systematic strategy. Fluorescent characterization showed good fluorescent properties of the hybrid polymer and the quench mechanism is investigated. As an additive to PET, the SiO2@Tb3+(PET-TEG)3Phen hybrid polymer was well blended with PET matrix to improve the Tc and Tm by 5.7% and 7.3%, respectively, with respect to pure PET. The improved crystallinity was ascribed to the portion of SiO2 nanoparticles, which function as nucleation sites. The Tb3+ acted as the fluorescent centre, and the PET-TEG segments played the role of linker and buffer to contribute better dispersibility of SiO2 nanoparticles in the PET matrix. Overall, this work paves a way for synthesis of multifunctional polymer hybrids to meet the needs in industrial and biomedical fields such as display screens and drug delivery indicator.

Round 3

Reviewer 1 Report

First, this review apologizes for the unavailability to review in the date indicated.
The present manuscript entitled "Fluorescent SiO2@Tb3+(PET-TEG)3Phen Hybrids for Enhancement of Crystallinity of PET, polymers-683915" includes enough information that improves the quality of the research. Also, the answers provided by the authors clarified the status of the manuscript. This new version can be accepted for publication in "polymers".